# The relative contributions of subjective and musical factors in music for sleep

Rory Kirk[1*], George Panoutsos[2], Maan van de Werken[3], Renee Timmers[1,4]

1 Muses, Mind, Machine, Department of Music, University of Sheffield, Sheffield, United Kingdom,
2 Department of Automatic Control and Systems Engineering, University of Sheffield, Sheffield, United Kingdom, 3 SleepCogni, BrainTrain2020 Ltd., Sheffield, United Kingdom, 4 Healthy Lifespan Institute, University of Sheffield, Sheffield, United Kingdom

* r.kirk@sheffield.ac.uk, rjdkirk@gmail.com

## Abstract

Previous research into music for sleep has focused on describing the types of musical characteristics associated with such music. The current study aimed to increase understanding of sleep music by investigating subjective perceptions of listeners associated with music that is considered sleep inducing. A listening experiment asked participants to rate musical excerpts along subjective evaluation dimensions related to valence, arousal, and dissociation. Musical features of the stimuli presented were extracted to compare the relative contribution of subjective and objective aspects. Our results reveal important roles for valence and arousal, and highlight notions of comfort, liking, and dissociation that contribute to music that is perceived to be sleep inducing. The musical analysis largely conformed with previous research, with an emphasis on brightness. However, the subjective ratings overshadowed the musical features in predicting what music was perceived to be most sleep inducing. Our findings are relevant for music recommendation applications that rely on a features-based approach to selecting music to fulfil particular purposes. As applications increasingly emphasise personalisation, we have shown that an account of subjective appraisals is crucial to predict listeners' experiences, providing recommendations for individually targeted therapeutic applications.

## Introduction

There has been a growing awareness of the proliferation of sleep problems in modern society and their profound impact on health and wellbeing [1]. Efforts are being made to improve awareness of sleep health and understand methods for alleviating sleep issues [2]. As one such method, the use of music to help with sleep has created a considerable amount of research attention. Studies have explored the efficacy of music as a sleep aid generally as well as in clinical settings with positive results [3,4]. However, the topic remains complex with a high degree of individual variability. Thus, continued efforts are needed to better understand how music can be used to help with sleep and optimise its potential for therapeutic application.

**Data availability statement:** All data are available from the data repository managed by the University of Sheffield, DOI: https://doi.org/10.15131/shef.data.25442095

**Funding:** RK was supported by a Doctoral Scholarship from the Engineering and Physical Sciences Research Council, UK Research and Innovation, award no. EP/R513313/1. The funders played no role in the conducted research or preparation of the manuscript.

**Competing interests:** MvdW is a co-founder and the Chief Scientific Officer at SleepCogni, Braintrain2020 Ltd. This does not alter the authors' adherence to PLOS ONE policies on sharing data and materials.

One such area that requires better understanding is the properties and qualities of music that can best promote sleep. In experimental studies researchers typically choose music that is described as relaxing, soothing, or sedative, with a slow tempo and little rhythmic or dynamic variation [5]. Many cite guidelines given by Gaston for so-called sedative music [6,7], and Nilsson's recommendations for the types of music suited for clinical settings [8,9]. However, individuals who report listening to music to help with their sleep describe a considerably more varied selection of music [10,11]. For example, Dickson & Schubert [10] found that music their participants identified as successful at aiding sleep was characterised by medium tempo, legato articulation, major mode, and the presence of lyrics, which they concluded did not conform to the usual sedative, instrumental music recommendations. For example, tempo was 107 bpm on average, compared to 52–85 bpm typically seen in sleep studies [5]. In an analysis of Spotify playlists, Scarratt et al. [12] found the most popular track across 985 playlists that were intended for sleep (i.e., had sleep in the title or description, a total of 225,626 songs) was Dynamite by Korean pop group BTS, an upbeat track filled with syncopation and a busy rhythm section. Their dataset overall leaned towards ambient and instrumental music, but was nonetheless distinctly diverse, highlighting the individual variation in the choice of music used to facilitate sleep [12].

These findings raise questions for how experimental research studies choose music for the purposes of aiding sleep. Indeed, the music used often comes from a range of sources or different selection processes, including music specifically composed for sleep [13,14], music chosen based on perceived levels of relaxation [15–17], or music selected by the participants themselves [18,19]. Some studies carry out preliminary work to validate their selections [13,20], however most do not report if any such preliminary work was conducted. This could mean the music selected is not in fact optimal. However, most studies tend to support the use of music as a sleep aid, possibly suggesting that music works as a more general intervention and requirements may not be so specific. Nonetheless, assessing the therapeutic and even clinical application of music requires a systematic understanding of the factors that contribute.

A promising avenue of investigation is to better understand the sleep-related affordances that music may offer to individuals and how these interact with acoustical and musical properties. In our study, we explore how music that is perceived to be sleep inducing is conceptualised along a number of subjective dimensions by listeners, including themes relating to valence, arousal, absorption, comfort, and liking. We also conduct a musical features analysis to complement our assessment and tie together the acoustical and perceptual properties of music for sleep. We hypothesise that it is the interaction between musical features and subjective factors that is predictive of music's potential to help with sleep.

### Psychological factors influencing sleep, and the potential of music

Sleep is a vitally important part of everyday life, yet a reported 30–40% of individuals suffer from at least one night-time insomnia symptom [21]. A range of factors can affect the quality of an individual's sleep, including environmental factors, such as

work requirements and lifestyle, or biological factors such genetics and overall health [1]. Some factors are external, while others relate to internal physiological and psychological mechanisms. As such, sleep problems are sometimes referred to as a disorder of hyperarousal, described by Levenson et al. [21] as heightened physiologic, affective, and/or cognitive activity. Psychological factors are seen to play a particular role in the sleep-wake cycle, where sleep onset is seemingly moderated by a cognitive signal that gives the body implicit 'permission' to relax and fall asleep, referred to as the 'lights out' effect [22]. This psychological switch can be difficult to control. Negative thoughts or other day-to-day physical and psychological stressors can adversely affect arousal and mood, preventing the body from reaching a sleep-ready state.

Regulating arousal and mood is considered to be one of the key functions of music listening generally [23], commonly used to relax and alleviate stress. Thus, music listening seems a suited method to help relieve these psychophysiological barriers and promote sleep onset. In light of this, Jespersen & Vuust [20] have proposed three mechanisms by which music can help with sleep. These relate to music's effects on physiological arousal, emotional effects, and as a means of distraction. Music that finds a balance between positive emotions, promoting mood and relaxation, and causes a decrease in sympathetic activity and an increase in parasympathetic activity would be expected to promote sleep. Further, music can act as a focal point of attention that distracts a listener from stressful thoughts [24], a suggestion that is supported by reports from individuals as a reason they use music to help with sleep [11].

These mechanisms overlap with general conceptions around the functions of music listening. Schäfer et al. [23] suggest that people listen to music to regulate arousal and mood, to achieve self-awareness, and as an expression of social relatedness. Each factor could be important for facilitating sleep. Regulating arousal and mood are specifically relevant to Jespersen & Vuust's [20] mechanisms. Achieving self-awareness may be relevant for certain individuals, for example by way of introspective contemplation (e.g., meditation) that may be a pathway to relaxation prior to going to sleep. The sense of social relatedness may also be valuable. Some theories suggest that music may improve wellbeing by providing a sense of belonging and connectedness in the absence of social interaction, acting as a social surrogate [25,26]. These associations may help to evoke a sense of comfort and security conducive to sleep. Indeed, the human thermoregulatory system, key to the sleep-wake cycle [27], is thought to be affected by feelings of social bonding whereby changes in body temperature are associated with feelings of social connectedness [28,29]. This may be linked to evolutionary developments that reinforced social behaviours in early humans [28,29]. Thermoregulation is key to the extent that temperature changes may have a causal effect on sleep by affecting particular neuronal activity that triggers sleep onset [27].

More broadly, the concepts of arousal and valence are central to the study of emotional affect. In sleep music studies, much of the arousal component is generally assumed, with sleep music typically expected to be lower in energy. Valence implications are arguably equally important for sleep music, however surprisingly few studies investigate this systematically and if analysed the picture is complex. Scarratt et al. [12] found that the valence measure provided by Spotify in their Data Catalogue was significantly lower, i.e., more negative, for tracks in sleep playlists compared to music from the Music Streaming Sessions Dataset, a publicly available dataset released by Spotify [30] consisting of audio features and metadata for approximately 3.7 million tracks, used by Scarratt et al. [12] to represent 'general' music. This finding appears to contradict the suggestions by Jespersen & Vuust [20] that sleep music should be positive. However, this result is difficult to interpret given the proprietary nature of the features in Spotify's Data Catalogue and the lack of details published regarding their underlying mechanics. Negative valence could be due to the interpretation of certain features such as lower pitch and subdued brightness [31]. In our current study, we can assess valence more directly from participants' ratings. We expect the valence factor to be relevant for positive promotion of sleep, countering the potential for depressive states also associated with low arousal.

Other affordances may also be relevant for music as a sleep aid. One such element is the notion that music can be used as a distraction [20], however this is difficult to qualify. Distractions can have negative consequences that prevent dissociation required for sleep; music could help to relieve certain distractions, or be the cause of negative distraction itself. Indeed, Dickson & Schubert [32] found that distraction was both a reason for and against using music for sleep,

for some "providing a blockage to […] negative thoughts" (32, p. 191) but for others stimulating too much concentration or triggering emotions that would hinder their sleep. For one participant, simply "any form of noise would distract me and keep me awake" (32, p. 191).

In this light, we may benefit from also considering the concepts of absorption and engagement. The terms are often conceptualised with relation to each other and associated with dissociation, but having different connotations with respects to consciousness in experience [33,34]. Similar to distraction, absorption and engagement may operate in complex and nuanced ways. Herbert [34] suggests that music "affords multiple entry points to involvement" (34, p. 57), including multiple potentially effective processes to sooth, relax and wind down. Music also facilitates an "altered relationship to self and environment" (33, p. 372). If the intention is to free the mind of stressful thoughts, more than to create a strong focal point, there may be a sweet spot where music can achieve this and facilitate sleep.

### Advancing existing research

The theoretical framework suggested by Jespersen & Vuust [20] provides a basis for empirical validation, namely for the effects on arousal, emotion, and distraction. In addition, we propose further concepts that are relevant for music listening and may play a role in sleep, such as comfort, engagement, and absorption. Given the overlap with generally proposed music listening functions [23], it is important to consider how these conceptualisations relate to sleep music more specifically by contrasting with music for other purposes. Such a comparison risks however comparing very contrasting types of music, observing more differences than required. To address this, we solicited the involvement of composers to create music specifically for the purpose of this study in addition to comparing features and conceptualisations of commercially available music.

### Current study

An experimental study was conducted to empirically investigate the subjective conceptualisations of sleep music. Additionally, we aimed to investigate the relative contribution of subjective qualities and objective musical features in the assessment of music as sleep inducing. By integrating subjective and objective analysis we can better elucidate the importance of these aspects and reflect on separate indications from previous literature. Our study is guided by the following research questions:

RQ1: What are the subjective qualities associated with sleep music?

RQ2: What are the objective musical features associated with sleep music?

RQ3: What is the relative contribution of subjective and musical factors in music perceived to be sleep inducing?

An online listening study was designed to gather ratings from listeners in response to a selection of musical pieces. To characterise responses to sleep music in relation to other forms of music listening, we included music suited for the purpose of sleep with music for relaxing and energising, as categorised by our selection process (see Methods section for details). Participants were asked to evaluate the music along 13 bipolar dimensions capturing subjective responses related to valence, arousal, comfort, engagement, absorption, and distraction. The MIR Toolbox for MATLAB [35,36] was used to extract a selection of musical features from the stimuli. Participants were also invited to leave comments during the study to provide further qualitative data.

## Materials and methods

### Ethics statement

This study received ethical approval from the University of Sheffield (application reference No. 036383). Data was collected between 15/03/2021-11/06/2021 and informed written consent was obtained from all participants through a consent form at the commencement of the survey.

## Stimuli

Stimuli consisted of 56 one-minute excerpts. Following a previous study of Spotify playlists [31], music was selected on the basis of falling into three categories: music for the purpose of sleep, music for relaxing, and music for energising. This included commercial music sampled from Spotify playlists, music gathered from previous sleep studies, and novel compositions created for the purposes of this study (see below for details). The intention was to create a diverse set of stimuli to draw sufficient comparisons. No specific genre of music or other such prerequisites were determined for any of the selections.

**Novel compositions.** Novel pieces of music were commissioned to gather material that has stylistic uniformity (is comparable across pieces from a single composer) whilst serving different purposes. MA students in Composition at the University of Sheffield were asked to compose a set of three pieces of music one minute in length suitable for the purposes of energising, relaxing, or sleep induction. The instruction was to compose excerpts that were closely related to each other, i.e., following a similar theme or base material, but varied in characteristics to differentiate between the different purposes. Eight composers returned a total of 24 pieces, including a variety of interpretations and stylistic contrasts (e.g., solo instrumentals and larger arrangements; acoustical pieces and electronic compositions).

**Selection from spotify.** To expand the selection of music, a matched number of tracks were selected from Spotify playlists that had been analysed in a previous investigation [31]. The original analysis concerned 4,500 tracks from Spotify playlists that were collected using search terms related to sleeping, relaxing, and energising. We used a Principal Component Analysis (PCA) on twelve audio features provided by the Spotify Data Catalogue to sample tracks deemed typically representative in each category based on component scores. See S1A Table for further details including a tracklist. The resulting 24 tracks consisted of mainly pop and dance songs in the energising playlists, all of which contained vocals, while most songs in the relaxing playlists were from the chill-hop genre, with only two songs in this set containing vocals. The sleep selection was entirely instrumental, consisting of mainly solo jazz and classical piano pieces.

**Commercial sleep music.** The selection of sleep music produced through our Spotify sampling process did not include ambient or meditative music such that is often prescribed in sleep studies, possibly reflecting the variety of music in public playlists [31]. To address this and add depth to our selection, a third set of sleep music was added to the musical materials to represent music purposefully composed to facilitate sleep and purposefully selected as music for sleep in research testing musical interventions.. A review of previous research resulted in a further eight tracks selected for this study. This track list and respective citations can be found in S1B Table. This selection will hereby be referred to as commercial sleep music (CSM).

**Sound file preparation.** Files for all 56 tracks were cut to the first minute with a three second fade out and exported as uncompressed 24-bit WAV files. We used YouTube to host the audio files online for embedding into the survey. Videos were created for each sample, set to a plain black background and exported to Standard Definition 480p .mov files. All uploads were set as Unlisted videos on purpose-created channels linked to the first author's University Google account, simply titled numerically from 1–56.

## Questionnaire items

**Background and mood questions.** Several background questions were presented in this survey, including musical engagement, music preferences [37], personality [38], sleep health and habits [39], and questions on self-help methods used for sleep and reasons why, compiled from previous surveys [11,40,41]. A full analysis of these items is outside the scope of the current article, which will focus on the music and subjective ratings.

Participants were asked to rate their mood, alertness, and tension before the listening phase along three 9-point bipolar scales intended to indicate valence, energy, and tension: Negative-Positive, Extremely Alert-Extremely Sleepy, Tense-Relaxed..

**Subjective responses to music.** Listeners were asked to rate each musical excerpt along 13 dimensions using 9-point bipolar scales. These were presented as ratings for describing the music and describing the effects of the music on the listener (see Table 1). Items related to emotional valence, energy and tension arousal, comfort, engagement, absorption, and distraction. Because distraction could be differently construed, it was presented with "freeing the mind" as its antithesis to convey the notion of dissociation, as opposed to attracting attention. Finally, an open comment box was provided for additional feedback.

## Musical features

A selection of musical features was extracted using the MIR Toolbox for MATLAB [35,36]. Our choice of features largely followed [10] for comparison, but with some additions to provide a more detailed analysis. As well as a measure of dynamic variation, we extracted the total dynamic energy of each track. Instead of an aural assessment of rhythmic activity, we included measures of event density and pulse clarity. As well as rhythmic and dynamic variation, we assessed modal variability by extracting key clarity. Brightness was also included following previous work that found this measure to be the strongest predictor for distinguishing sleep and relaxing playlists from Spotify [31]. The full list of features and their descriptions can be seen in Table 2.

## Procedure

The survey was conducted online using the Smart Survey platform and disseminated through University of Sheffield mailing lists and other online platforms, including Facebook, Reddit, and Twitter. No specific demographic was targeted, and the only requirement was that participants had no hearing impairments. Full information on the study and a consent form was provided at the beginning of the questionnaire. Each participant was presented with a random subset of 14 examples from the 56-piece selection, balanced across each category (i.e., two excerpts from each main category (sleeping, relaxing, energising) and source (Spotify, novel compositions) combination plus an additional two CSM pieces). Participation was completely voluntary and participants were not rewarded for completing the study.

**Table 1. Listening phase ratings questions. Participants were asked to indicate in one direction or another along a 9-point scale.**

| I would describe the music as: | | |
|---|---|---|
| | Negative | Positive |
| | Tense | Relaxed |
| | Sleepy | Awake |
| | Familiar | Unfamiliar |
| | Boring, unappealing | Engaging |
| The effect of the music on me can be described as: | | |
| | Pleasant | Unpleasant |
| | Calming | Activating |
| | Energising | Sedating |
| | Comforting | Distressing |
| | Absorbing | Repelling |
| | Distracting | Freeing the mind |
| | Sleep inducing | Sleep preventing |
| How much do you like/dislike the music? | | |
| | Like | Dislike |

**Table 2. Musical features included in our analysis. Includes features assessed by Dickson & Schubert, 2020, hereby referred to as D&S.**

| Feature | Description | Type of analysis |
|---|---|---|
| Articulation | Mean ratio of the decay in amplitude over time. | MIR Toolbox using the Decay Slope Mean, following D&S. |
| Brightness | Level of upper mid and high frequency content. | MIR Toolbox using the mirbrightness function. D&S measured the mean frequency spectrum centroid (Hertz), which they compare to brightness. |
| Dynamic energy | Global energy of the signal using the root mean square (RMS) amplitude. | MIR Toolbox using the mirrms command. |
| Dynamic variation | Standard Deviation of the root mean square (RMS) amplitude. | MIR Toolbox using the mirrms command, following D&S. |
| Event density | Average frequency of events per second. | MIR Toolbox using the mireventdensity command. |
| Key clarity | The key strength associated with the best estimation of the tonal centre. | MIR Toolbox using the mirkey command. |
| Mode | Major vs minor. | MIR Toolbox using mirmode function, which returns a value between +/-1 to indicate the degree of major/minor mode. D&S used aural analysis (Major/Minor). |
| Pulse clarity | Estimates the rhythmic clarity, indicating the strength of the beats. | MIR Toolbox using mirpulseclarity command (Lartillot, Eerola, et al., 2008). |
| Tempo | Calculation of beats per minute (bpm) | MIR Toolbox using the mirtempo command. D&S chose manual tempo tapping (Tap BPM). |

Participants were asked to carry out the listening portion in the evening to be more appropriate for the topic of the study, considering circadian effects that may affect mood and alertness [42]. However, due to the inconvenience this may cause, limitations on recruitment, and challenge to reinforce, it was not considered as a strict requirement. To provide a reflective assessment of the adherence to this request, we used the finish times recorded by Smart Survey as a proxy. Participants whose finish times were at least 30 mins after 6 pm in their respective time-zone were considered to have completed the listening phase in the evening. This assessment suggested that 62 (57%) of the participants did complete the study in the evening. As this is only a portion of the sample, this requirement was not met, and we will consider this to be a limitation of our study. A debriefing page was included at the end of the survey with an open comment box for participants to provide extra feedback. According to the timings recorded by Smart Survey, the survey took approximately 20–40 minutes to complete.

## Analysis

First, we investigate how the music categories differ in terms of ratings and musical features to address RQs 1 and 2. For this comparison, ratings were averaged within each category by participant. One-way repeated measures ANOVAs were used to compare ratings between the music categories followed by paired samples t-test with Bonferroni corrections for post hoc pairwise comparisons. Kruskal-Wallis H tests were used to compare the musical features between music categories due to non-normal distribution of the data, as assessed by visual inspection of histograms and confirmed by Shapiro-Wilks tests, with pairwise comparisons performed using Dunn's procedure [43].

Next, we usedPCA on the full ratings dataset including all musical tracks and individual responses (not averaged within categories) to explore underlying patterns in the ratings. It was assumed that there may be strong relationships between several of the ratings dimensions, and by reducing the variables to their fundamental components we could investigate any further patterns in the data.

Finally, multiple linear regression was used to address RQ3 and investigate the contribution of combinations of all our variables to predict what makes a piece of music sleep inducing.

Further details on all analyses are provided in the Results section below and in S3 Table. All statistical analysis was carried out using SPSS 27, and Laerd Statistics (https://statistics.laerd.com/) was used for guidance on procedure and reporting.

## Results

### Participants

We received 108 complete responses (69 female (63%), 45% aged 21–29). Most participants spent their formative years in Europe (N = 79, 18 Asia, 11 other). 78 participants (72%) reported that they played or had played a musical instrument. Our sample had a mean Global PSQI score of 6.64, indicating generally poor sleep, on a scale from 0–21 where scores above 5 suggest significant sleep difficulties. 35 participants (32%) reported that they never used music to go to sleep (26 sometimes, but rarely; 28 occasionally; 13 quite often; 6 always). Further participant information can be found in S2 Table.

### Overview of ratings and features

The final dataset consisted of 1,512 entries (108 participants x 14 tracks each). Each track received 15–43 responses (*M* = 27, *SD* = 5.41). Distribution of ratings by music categories can be seen in Fig 1. For consistency and to aid interpretation, we have ordered each rating along its relative positive-negative valence or high-low arousal directionality. For example, Sleep Preventing is considered the high end as an arousal dimension, whereas Sleep Inducing is low. All figure and table labels correspond to the positive- or high- directed adjective of each dimension.

Repeated measures ANOVAs revealed significant differences between the categories for all of the ratings except Positivity and Familiarity and significant differences for many pairwise comparisons (see Fig 1 and S3A Table). Overall, music in the Sleep category was rated significantly less awake, activating, energising, distressing, and sleep preventing

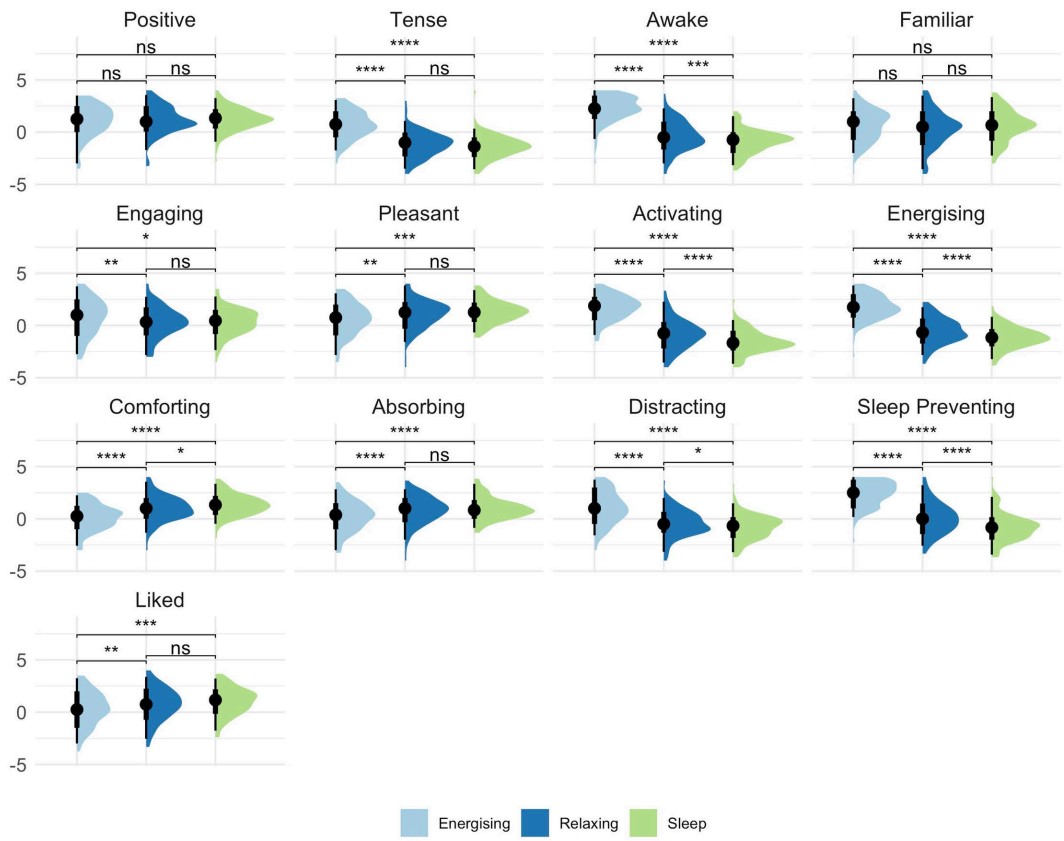

**Fig 1. Plots of all ratings by music category.** Results of paired samples t-tests are shown. ns = not significant. *p < .05. **p < .01. ***p < .001.

(i.e., more sleepy, calming, sedating, comforting, and sleep inducing) than Relaxing music, and additionally less tense and more pleasant, absorbing, and liked than Energising music. The same significant differences were found for Relaxing music as Sleep compared to Energising music.

Musical feature distributions per music category can be seen in Fig 2. Kruskal-Wallis H tests revealed significant differences for all features except Tempo and Mode (see S3B Table). For the remaining features, pairwise comparisons found significant differences between Energising and Sleep music for all but Dynamic Variation and Articulation. The latter was significant between Sleep and Relaxing music ($p = .018$). No other feature was significantly different between Sleep and Relaxing music. Relaxing music only differed significantly from Energising music for Pulse Clarity ($p = .001$) and Event Density ($p = .001$).

### Interrelations between subjective ratings

Linear analysis was run to investigate relationships between the different subjective dimensions. Overall ratings (i.e., not separated by playlist category) were close to normally distributed, as assessed by visual inspection of histograms and Normal Q-Q plots without strong outliers. Pearson correlation results (see S3C Table) are depicted in the heatmap shown in Fig 3. Many variables were highly linearly correlated. Correlations showed a clustering of variables into two distinct groups akin to positive-negative valence and high-low arousal, respectively.

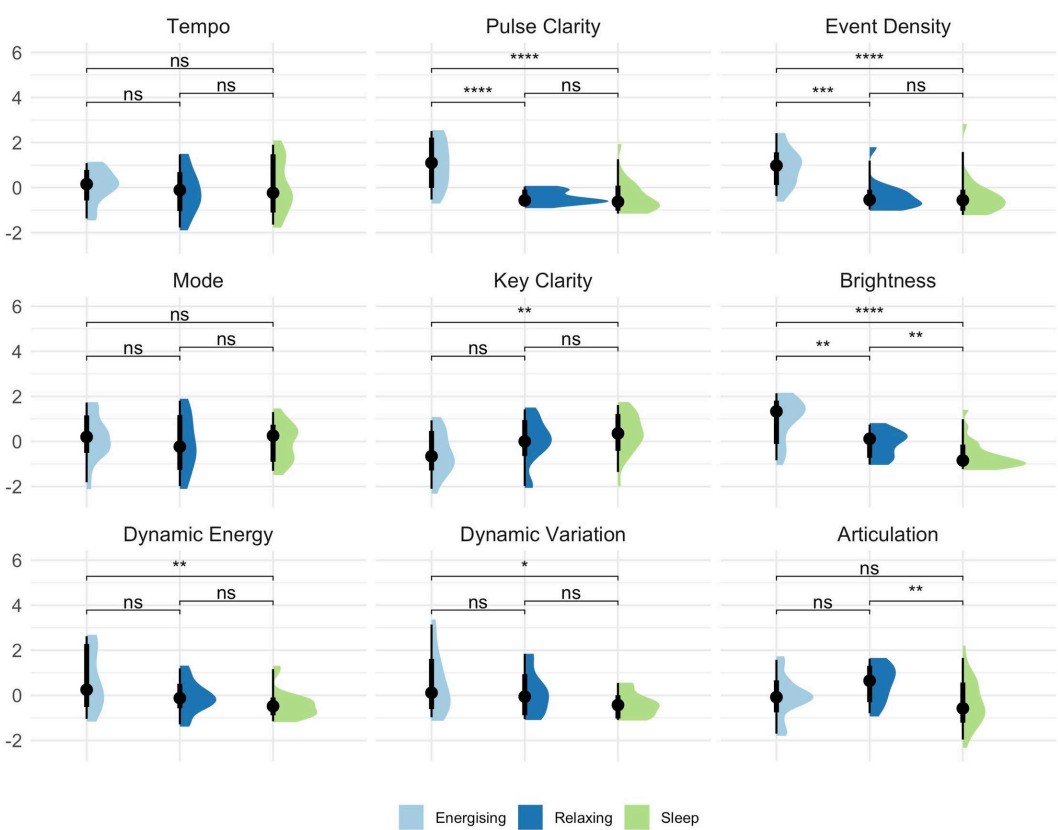

**Fig 2. Plots of musical features, excluding outlier tracks revealed during the regression analysis (see Results – Predicting sleep induction – subjective ratings and musical features).** All values are standardised. Results of Wilcoxon signed-ranks tests are shown. ns = not significant. *p < .05. **p < .01. ***p < .001.

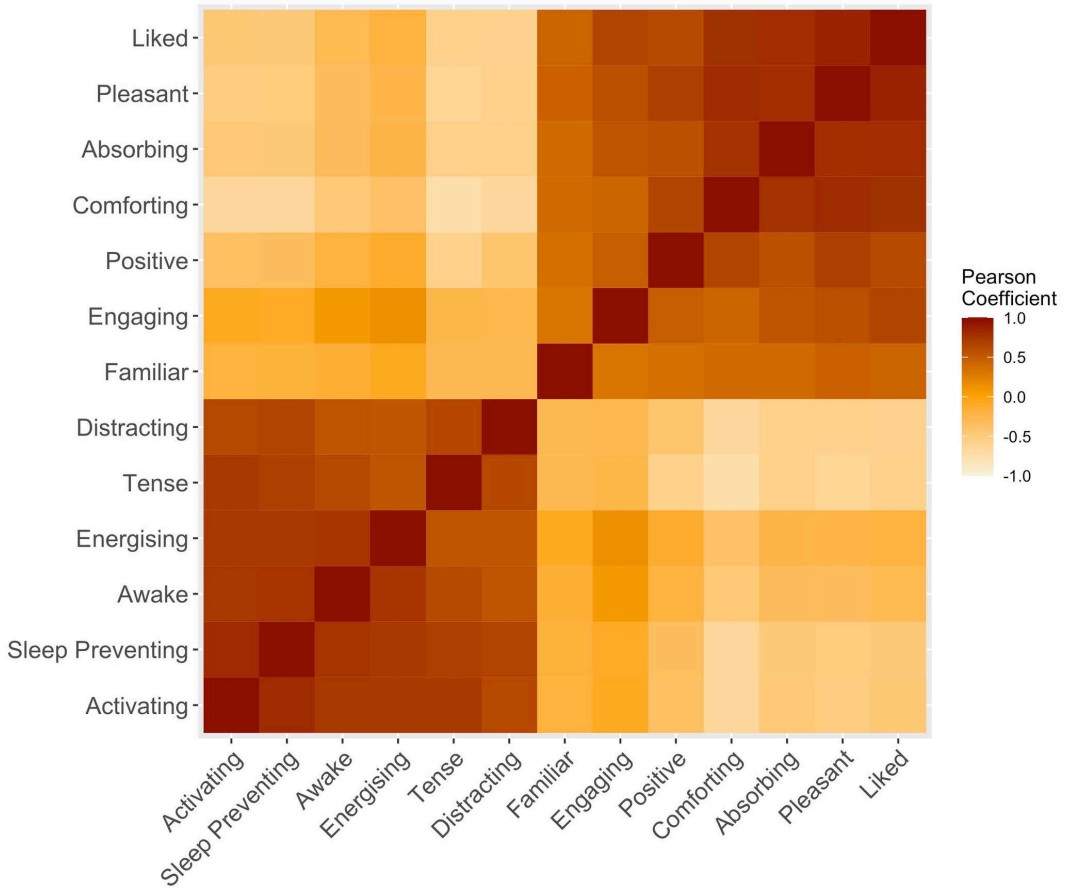

**Fig 3. Correlations heatmap showing correlation coefficients between pairs of evaluative dimensions ordered with hierarchical clustering.**

PCA was used to further examine this clustering. Suitability for PCA was first assessed by inspection of the correlation matrix (see S3C Table). All variables returned correlation coefficients greater than.3, and most were greater than.6 except Familiar-Unfamiliar (greatest.443). Familiar-Unfamiliar was also the only variable with a Communality coefficient less than.5 (.297). As a middling factor that did not fit as strongly with the other variables, we decided to rerun the analysis excluding Familiar-Unfamiliar. The resultant analysis gave an overall Kaiser-Meyer-Olkin (KMO) measure of.932, or "marvellous" according to Kaiser's classifications [44]. Bartlett's Test of Sphericity was significant ($p < .0005$), indicating that the data was likely factorisable. The PCA revealed two components with Eigenvalues greater than 1 explaining 77.2% of the total variance. Varimax rotation (Table 3) revealed a close to simple solution, with most factors loading exclusively on one or the other component. The first component contained variables which might relate to a listener's positive experience of the music, whereas the second contained activation factors, befitting a valence-arousal distinction. From that perspective, we can also see a possible tension arousal overlap with Tense-Relaxed and Comforting-Distressing falling into both components, reminiscent of a three-factor model [45,46]. A forced three-component extraction did not reveal a tension dimension, instead resulting in a more complex solution with no logical interpretation of the components. We therefore retained the two-component solution and will hereafter refer to these components as the Valence component and the Arousal component. The Valence component accounted for a larger proportion of the variance in responses (57%) than the Arousal component (20%).

**Table 3. Varimax rotated component matrix, final solution. Coefficients < .3 are suppressed.**

| Rating | Valence Component (56.9%) | Arousal Component (20.3%) |
|---|---|---|
| Liked | .888 | |
| Pleasant | .878 | |
| Absorbing | .829 | |
| Engaging | .802 | |
| Comforting | .764 | −.509 |
| Positive | .760 | |
| Energising | | .897 |
| Awake | | .895 |
| Activating | | .868 |
| Sleep Preventing | | .867 |
| Tense | −.503 | .697 |
| Distracting | −.466 | .650 |
| Method: Principal Component Analysis | | |
| Rotation: Varimax with Kaiser Normalisation | | |

Component scores were used to explore the dimensional structure of the music selection. Average component scores were calculated for each track and are presented in Fig 4, separated by the three categories (Sleep, Relaxing, Energising). There is some separation of each category, with the Sleep and Relaxing music occupying a similar space lower than the Energising music on the arousal scale. There appear to be linear trends in different directions between the Sleep and Relaxing music compared to the Energising music. For Sleep and Relaxing music, Arousal values decrease with increased Valence, while the opposite is the case for Energising music. Kendall's tau correlation analysis for skewed data found this relationship between Valence and Arousal was only significant in the Sleep category (N = 23, τ = −.383, p = .010) but not for Relaxing (N = 15, τ = −.352, p = .074) or Energising music (N = 16, τ = .200, p = .310).

### Predicting sleep induction – subjective ratings and musical features

To assess which qualities correspond to how sleep inducing a piece of music is perceived to be, we used the Sleep Preventing-Sleep Inducing rating as a dependent measure in standard multiple linear regression models of the ratings and acoustic features.

Analysis of the ratings revealed potential collinearity issues based on an assessment of Variance Inflation Factors (VIF). Pleasant-Unpleasant, Comforting-Distressing, and Like-Dislike each had VIF values >5. Pleasant was the only variable with correlation coefficients > .8, with both Comforting-Distressing and Like-Dislike, so this was removed. Comforting-Distressing still had a marginally high VIF value (5.151) in the subsequent analysis, so we will review this outcome with some caution. All other assumptions were satisfied. This model statistically significantly predicted the sleep induction ratings, $F(11, 1435) = 443.955$, $p < .001$, adj. $R^2 = .771$, and several of the ratings added statistically significantly to the prediction (see Table 4, Model A). The highest coefficients were returned for the variables associated with Arousal. Of the Valence variables, Comforting-Distressing had the highest value, above liking, freeing of the mind, and familiarity. None of the other Valence-associated variables significantly contributed to the model.

Next, we looked at how well the musical features predicted the Sleep Preventing-Sleep Inducing rating. The first analysis found collinearity issues with the two RMS outputs (Dynamic Energy and Variation); therefore, the analysis was rerun with only Dynamic Variation, keeping in step with Dickson & Schubert [10]. In the second attempt, two tracks returned Leverage values greater than.5, and these were subsequently removed. The final model statistically significantly predicted

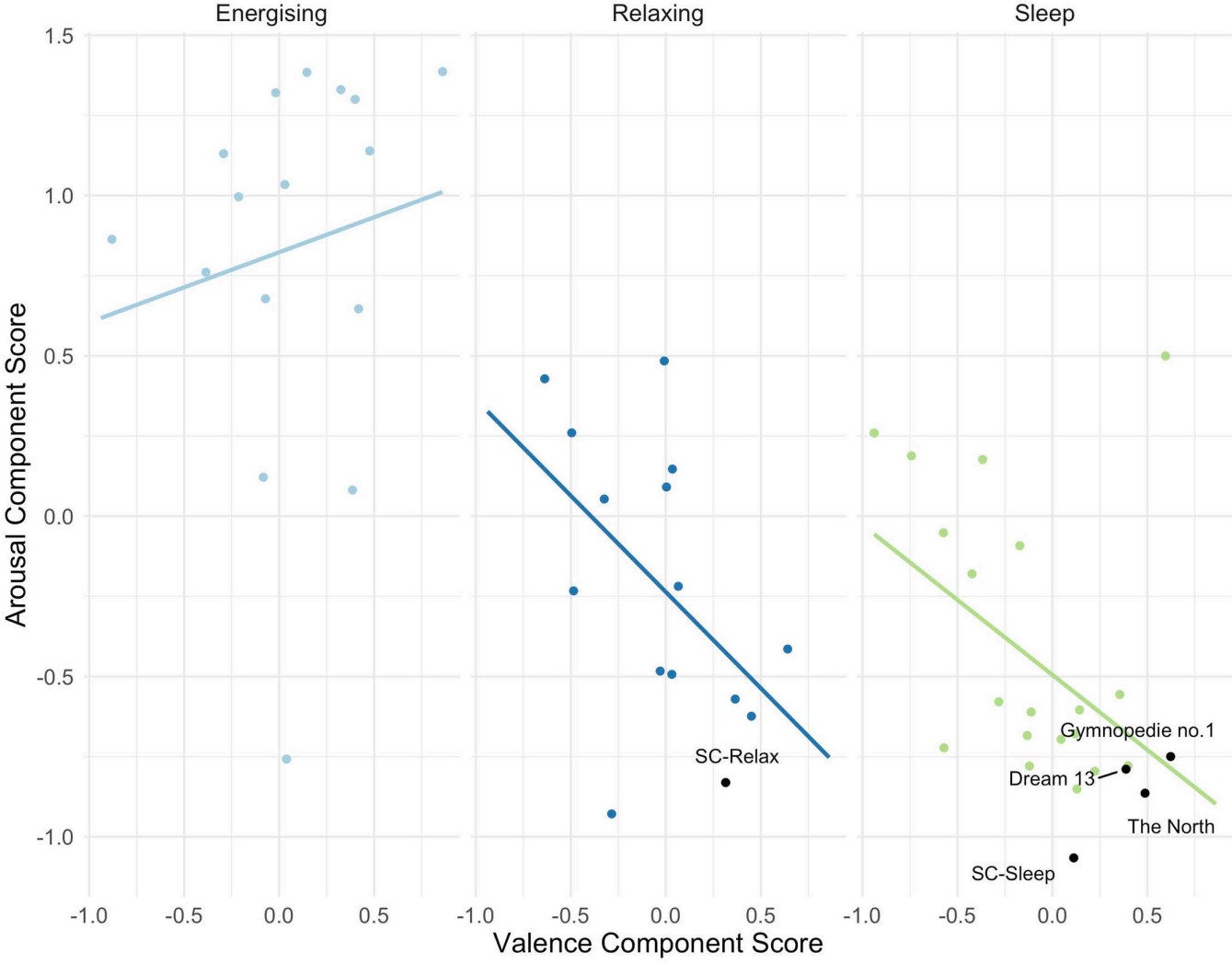

**Fig 4. Average component scores for each track by playlist category with trend lines between Valence (x-axis) and Arousal (y-axis) component score.** The five tracks with the lowest average Sleep Preventing ratings (i.e., were rated as highly sleep inducing) are labelled. These concern two tracks by one composer at the University of Sheffield, labelled as SC, Dream 13 (minus even) by Max Richter, The North by Niels Eje, and Gynopédie No. 1 by Eric Satie.

the sleep induction ratings, $F(8, 1438) = 89.110$, $p < .001$, adj. $R^2 = .328$, and Brightness, Event Density, and Pulse Clarity added statistically significantly to the prediction (see Table 4, Model B).

Finally, a combined analysis was carried out including all of the variables above. The model statistically significantly predicted the sleep induction ratings, $F(19, 1427) = 258.981$, $p < .001$, adj. $R^2 = .772$ (Table 4, Model C). The same subjective variables were significant as in Model A. Fewer musical features were significant: Brightness ($p = .859$) and Pulse Clarity ($p = .598$), were no longer statistically significant; instead, Dynamic Variation was found to significantly predict sleep induction ratings ($p = .026$) in addition to Event Density ($p = .045$).

## A closer look at the most sleep-inducing pieces

To tie together our analyses and help interpret the results, we took a closer look at the five tracks with the highest ratings towards sleep-inducing, as determined by mean ratings (highlighted in Fig 4. A top ten list can be found in S4 Table). Two

**Table 4. Multiple regression results for Sleep Inducing-Sleep Preventing by ratings and musical features separately, then together. Factors are ordered by significance and standardised beta coefficient values. For Model C, ratings are reported first, followed by the features.**

| Sleep Preventing | B | 95% CI for B | | SE B | ß | Sig. | $R^2$ | $\Delta R^2$ |
|---|---|---|---|---|---|---|---|---|
| | | LL | UL | | | | | |
| Model A | | | | | | | .773 | .771 |
| (Constant) | .656 | .577 | .735 | .04 | | .000 | | |
| Activating | .298 | .247 | .349 | .026 | .293 | .000 | | |
| Awake | .266 | .222 | .311 | .023 | .26 | .000 | | |
| Energising | .204 | .153 | .254 | .026 | .178 | .000 | | |
| Comforting | −.202 | −.271 | −.134 | .035 | −.166 | .000 | | |
| Liked | −.106 | −.161 | −.051 | .028 | −.101 | .000 | | |
| Distracting | .1 | .058 | .141 | .021 | .09 | .000 | | |
| Familiar | .061 | .031 | .09 | .015 | .057 | .000 | | |
| Positive | .035 | −.01 | .08 | .023 | .028 | .124 | | |
| Engaging | .028 | −.013 | .069 | .021 | .024 | .185 | | |
| Tense | .012 | −.037 | .06 | .025 | .011 | .643 | | |
| Absorbing | −.006 | −.063 | .05 | .029 | −.005 | .821 | | |
| Model B | | | | | | | .331 | .328 |
| (Constant) | −1.316 | −2.546 | −.085 | .627 | | .036 | | |
| Brightness | 3.523 | 2.572 | 4.473 | .485 | .271 | .000 | | |
| Event Density | .784 | .594 | .975 | .097 | .265 | .000 | | |
| Pulse Clarity | 1.574 | .914 | 2.234 | .336 | .139 | .000 | | |
| Articulation | .058 | −.005 | .12 | .032 | .042 | .070 | | |
| Mode | −.639 | −1.643 | .366 | .512 | −.029 | .212 | | |
| Key Clarity | −.709 | −2.322 | .903 | .822 | −.023 | .388 | | |
| Tempo | 0 | −.003 | .003 | .002 | −.003 | .883 | | |
| Dynamic Variation | −.204 | −4.981 | 4.573 | 2.435 | −.002 | .933 | | |
| Model C | | | | | | | .775 | .772 |
| (Constant) | .558 | −.169 | 1.284 | .37 | | .132 | | |
| Activating | .281 | .228 | .333 | .027 | .276 | .000 | | |
| Awake | .263 | .217 | .308 | .023 | .257 | .000 | | |
| Comforting | −.212 | −.28 | −.144 | .035 | −.174 | .000 | | |
| Energising | .195 | .145 | .246 | .026 | .171 | .000 | | |
| Liked | −.111 | −.167 | −.056 | .028 | −.106 | .000 | | |
| Distracting | .096 | .055 | .138 | .021 | .087 | .000 | | |
| Familiar | .057 | .027 | .087 | .015 | .054 | .000 | | |
| Engaging | .032 | −.01 | .073 | .021 | .028 | .135 | | |
| Positive | .026 | −.02 | .071 | .023 | .021 | .273 | | |
| Tense | .01 | −.04 | .059 | .025 | .009 | .698 | | |
| Absorbing | −.001 | −.057 | .055 | .029 | −.001 | .980 | | |
| Dynamic Variation | 3.223 | .379 | 6.067 | 1.45 | .035 | .026 | | |
| Event Density | .118 | .003 | .233 | .059 | .040 | .045 | | |
| Tempo | −.002 | −.004 | 0 | .001 | −.025 | .055 | | |
| Mode | .576 | −.02 | 1.172 | .304 | .026 | .058 | | |
| Pulse Clarity | −.106 | −.501 | .288 | .201 | −.009 | .598 | | |
| Articulation | .007 | −.029 | .044 | .019 | .005 | .694 | | |
| Brightness | −.052 | −.628 | .524 | .294 | −.004 | .859 | | |
| Key Clarity | .013 | −.935 | .962 | .484 | 0 | .978 | | |

*Note.* Model = "Enter" method in SPSS Statistics; *B* unstandardised regression coefficient; *CI* = confidence interval; *LL* = lower limit; *UL* = upper limit; *SE B* = standard error of the coefficient; *ß* = standardised coefficient; *$R^2$* = coefficient of determination; *$\Delta R^2$* = adjusted $R^2$.

of these tracks were written by a composer studying at the University of Sheffield. Of the remaining three, two were from commercial recordings that had been used in previous sleep studies, specifically selections from the albums Sleep by Max Richter [47] and MusiCure by Niels Eje [20]. The final piece is a recording of Erik Satie's Gymnopédie No. 1, a piece that appears in several studies as an example of relaxing music or reported by listeners as a piece used for sleep [11,48–50]. This track was rated overall as the most familiar, which could correspond to its greater average values for liking, pleasantness, comfort, absorption, and engagement, culminating in the highest Valence score of this selection. By contrast, the highest sleep-inducing rated piece had the lowest Valence score of these five, but also the lowest Arousal score, and indeed the lowest scores for most of the variables that contributed to this component (Awake-Sleepy, Energising-Sedating, Sleep Preventing-Sleep Inducing). The musical features of this piece were indeed maximally associated with lower activation; this piece had lower values for Articulation, Dynamic Energy and Variation, Event Density, and no clear pulse compared to the other five. Conversely, the track was more minor in Mode and had the highest Brightness values of the five.

Most of these pieces received very little in the way of comments from participants. Many other pieces received plenty of commentary, predominantly critical, personal, or expressions of enthusiasm in the case of the Energising pieces. The exception was Gymnopédie No. 1, again perhaps due to its familiarity. One participant exclaimed, "Erik Satie is one of my favourite composers:)" (P72), another stated that the piece was one of their "favourite pieces of music" (P82). Another discussed listening to the piece to help with their sleep during the SARS-CoV-2 pandemic:

When quarantine started I barely slept due to stress and anxiety and I used this piece to fall asleep for a month straight, it's more than comforting that song just feels like home. (P71)

The theme of comfort is echoed by another participant who described the piece as "very melancholic and comforting, it's like a cuddle for your soul" (P115). For others, however, the familiarity was not conducive to sleep, with one participant stating, "I love Satie, so this would keep me awake as I tried to remember the fingering!" (P85), and for another, "Individual notes are too distinct and the melody too familiar, would not allow me to disassociate" (P108). One participant elaborated further:

It only doesn't "free my mind" completely because I recognize the tune and as a musician I was predicting what came next while I listened. (This is why I can't listen to music to fall asleep!) Otherwise the piece itself, the same dynamic, the single timbre, and steady tempo... all were very relaxing. (P77)

## Discussion

In this study we investigated subjective associations and musical features to assess what characteristics of music may make it 'sleep inducing'. We used music from three sources (Spotify playlists, commercial sleep music, and novel compositions) that fell into three categories (Sleep, Relaxing, and Energising) to produce a broad selection for comparison. The results showed that the experience of music as sleep inducing was dependent on a combination of valence and arousal evaluations. Higher valence and lower arousal corresponded with greater experience of sleep induction. We found a prominent role for notions of comfort, which significantly predicted ratings towards sleep induction along with liking and freeing the mind, and was highlighted in participants' comments. Brightness, Event Density, Pulse Clarity, and Dynamic Variation were significant musical characteristics, however their relative contributions were small and differed between analyses. Overall, our results indicate that subjective appraisals are strong predictors of evaluations of music as sleep inducing, overshadowing musical attributes in predictive ability, and accounting for a high proportion of variance. In the following we discuss these results in more detail.

### Arousal and valence distinctions

Sleep Prevention was better predicted by variables associated with arousal, whereas valence was less prominent in our regression models. This could be a reflection of the music selection; although no variables were explicitly manipulated, the

categorical selection itself (from Energising to Relaxing to Sleep) connotes a dimension of arousal. However, the divergent trends seen in our PCA analysis still offer some intrigue. Higher Valence component scores seem to correspond to lower Arousal component scores for Sleep and Relaxing music, whereas Arousal scores were higher with increasing Valence for Energising music (see Fige 4). This association was only statistically significant for the Sleep music category, which contained a greater number of tracks (N = 24) in what was otherwise a small sample (N = 56). The orthogonal relationship (or possible lack thereof) between valence and arousal is complex [51], but the divergence seen here could indicate an interaction that differs depending on the goals of the music. For Energising music, the perception of greater arousal is enhanced when the music is enjoyed or deemed positive, whereas for Relaxing and Sleep music, where the intention is to reduce arousal, this is also better achieved when the music is seen as more positive. Crucially, we saw that the pieces rated as the most sleep inducing on average occupied the extreme end of the right lower quadrant of this space (low arousal, positive valence).

Our valence interpretation should remain loose. These patterns could also be a factor of the dimensions that feed into the component scores, such as engagement, absorption, and tension, which could have different relevance for Energising music compared to the other categories. Likewise, this could explain their lack of significance in our regression analysis. Explicitly manipulating valence would expand on these results. Nonetheless, we find support for the suggestion that positive valence is important for music for sleep, and may vary in association with arousal, both contributing towards the potential for sleep induction. This gives empirical support to the suggestions put forward by Jespersen & Vuust [20], that music best for sleep should be positive and low in arousal.

### Comfort and freeing the mind

As predicted, comfort was a significant factor for the experience of sleep induction, both predicting ratings and as a theme specifically referred to by participants in comments. Supporting feelings of comfort and safety may be important avenues by which music benefits wellbeing, possibly by acting as a social surrogate [25,26], and this may be another avenue by which music helps sleep. Although we haven't explicitly studied what makes a piece of music comforting this would be a fruitful avenue of further investigation.

Distraction is the third mechanism suggested by Jespersen & Vuust [20], and this was more difficult to distinguish. The Distraction-Freeing the mind rating significantly predicted sleep induction ratings, however ratings for absorption and engagement did not offer further insight as we had expected. Absorption was significantly negatively correlated with Sleep Preventing ($r = -.480$, $p < .001$), however neither Absorption nor Engagement were significant predictors in our regression analysis, possibly due to having different relevance for the different categories of music or correlation with other variables. Comparisons between music categories did find that music in the Sleep and Relaxing categories was significantly less Engaging and significantly more Absorbing than Energising music (see Fig 1). The significance of Freeing the mind suggests that dissociation is an important factor in music for sleep and needs further dissection.

### Familiarity

Familiarity is often considered influential in the context of music listening for mood regulation and emotional affect [52,53]. Previous surveys have highlighted the importance of familiarity for the listener when it comes to choosing music for sleep [10,11], and in our study participant comments highlighted both positive and negative aspects of familiarity. In some cases, the desire to predict or remember a piece as it unfolds was described as a hindrance. Indeed, familiarity was significantly negatively associated with sleep induction, according to our regression analysis. It is difficult to consolidate familiarity and the effect of predictability in this sense; for some, increased predictability might reduce attentional demand, and therefore cognitive effort, whereas the anticipation in listening to something novel might have the opposite effect and prevent dissociation. Our results offer both positive and negative associations with familiarity, with specific reference to predictability by participants, suggesting that this is a more complex relationship. It is possible that personal differences play a

role; individuals may have different requirements when it comes to their sleep and different cognitive approaches to music listening that place a variable role on familiarity and predictability. Musicality may also be a factor, with some participants' comments suggesting an influence of their instrumental musicianship. Furthermore, there could be indirect influences of familiarity through interactions with affective evaluations that influence the experience of music as sleep inducing. Clearly an important issue, a more explicit testing of familiarity will benefit future research on sleep music, and could explore potential implications for distraction, as well as the relationship with comfort, safety, and liking.

### Subjective vs. musical properties

Brightness was a significant predictor in one of our analyses, corroborating with previous work [31] indicating an important feature that is often overlooked in discussions around sleep music. It confirms findings of Dickson & Schubert [10] who found that music their participants reported as successful in helping with sleep was associated with lower main frequency register compared to unsuccessful music. Brightness can reflect different factors such as instrumentation, recording quality, or pitch, so it only provides a general indication of the timbral qualities of a track. Our results suggest that this is an important factor that should be investigated in future research.

Other significant features, Pulse Clarity, Event Density, and Dynamic Variation, point to rhythmic and dynamic aspects that are more commonly discussed in the sleep music literature, with our results aligning with the general assumptions of researchers [5]. Other indications further align with prior notions of the types and characteristics of sleep music; many of the highest sleep-inducing rated pieces, based on average responses, were piano based, soft, calm, and minimal in variation and event density.

Overall, our regression models indicate a greater importance of subjective evaluations for predicting what music was experienced as sleep inducing. Not only were the subjective factors more significant in the combined model (Table 4, Model C) but the accuracy of both Models A ($\Delta R^2 = .771$) and C ($\Delta R^2 = .772$) was considerably greater than the features only Model B ($\Delta R^2 = .328$). Although there were general trends in the musical characteristics of the highest sleep-inducing rated pieces, there was considerable individual variability that the features alone could not account for.

### Mechanisms

In our investigation we were particularly interested in the subjective experience of music which may support sleep induction, describing the music as it is perceived along dimensions related to arousal, valence, and supporting dissociation [20]. However, there can be a variety of reasons why individuals use music to help with their sleep that may interact with choices and experiences. Our investigation was primarily focused on the affordances of music; the potential to affect arousal or influence cognition. The latter concerns the goals or situation of the listener, which may emphasise the role of particular affordances over others. These concepts do overlap; Dickson & Schubert [32] and Trahan et al. [11] also highlight themes related to mood and distraction in their investigations on the reasons why people use music to help with sleep. However, the distinction is important to make as it pertains to our understanding of this complex area. Our investigation explored the subjective properties of music associated with sleep induction, not necessarily characterising the uses of music for sleep. The requirement to find a distraction may be particular to the circumstances of an individual; music affords one possible tool for addressing that need. A better understanding of this intersection between the affordances of music, the goals and uses of music, and the attentional demands of the music for listeners will be an important next step.

Masking is another common goal in listening to music for sleep, where music is used to cover up environmental sounds. In this sense, music may be alleviating one distraction by introducing another, or, seen another way, creating a sonic environment that reduces the effect of a negative distraction allowing for a more sleep-inductive frame of mind. Other tools may be applicable to achieve this goal as well. Listening to podcasts, for example, is another popular method used for going to sleep that is similarly a passive listening activity that could provide a form of distraction. A question

therefore remains as to why an individual might choose music over another method. This could be due to particular situational needs or constraints, or additional affordances. Like music, podcasts or audiobooks could contribute to creating a sense of comfort by way of providing a social surrogate [25], however music might be more conducive to positive moods for some individuals. Alternatively, the human voice may require different attentional demands [54] which could make it more effective than music for some individuals. Expanding our understanding of the use of music for sleep by comparing with alternative methods would be a valuable avenue for future investigations.

Our assessment was particularly focused on the affordances of music for sleep and was not an investigation of individuals using music for sleep. Indeed, most of our participants reported never or rarely using music to go to sleep themselves. This is potentially a limitation in our investigation, raising a question of how experiences in a listening study may translate to real experiences at night-time. Our results need to be validated in a real-world assessment, taking into account context driven and situated aspects of music listening that are important for sleep.

### Limitations

The musical selection process included music from Spotify playlists that were selected based on a best fit approach (see S1A Table). Given the extreme variety found in Spotify sleep playlists [12], our selection may be limited and is not guaranteed to represent what might be best for sleep. The CSM selection attempted to compensate for this and offered some expansion to our sleep music selection, but results for the category may still be limited. Potential music choices could be near inexhaustible, and a practical assessment requires that such selective processes are made. Nevertheless, an expansion of this work could look to further selections of different music, potentially explicitly manipulating certain parameters such as valence, arousal, or musical properties.

Our study used convenience sampling and our sample demographic was notably skewed. As a result we have not factored any participant background information into our analysis. Our focus was to explore general perceptions of a musical selection, but given the individual variability in responses a consideration of personal differences would be extremely valuable. For example, Lee-Harris et al. [55] found that for younger people relaxation was most strongly associated with levels of arousal, while for older people it was more associated with valence. Cultural context could also be extremely relevant in this case; singing lullabies to infants is a particularly unique universal musical activity with similarities in musical properties noted across cultures [56,57]. However, whether and how this may diverge in adulthood is less understood. Similarly, a more diverse musical sample may be important to capture diverging associations with music and sleep across cultures.

In this experiment, we used a number of dimensions that are new to this area of study, notably Boring-Engaging, Absorbing-Repelling and Distraction-Freeing the mind. Further validation of the reliability and validity of these dimensions is needed to make these dimensions into a validated questionnaire. In particular, we chose to consider distraction as a negative in our scale, but there could be other connotations that need to be captured. A complementary scale, or differentiating between positively and negatively distracting, may help to discern this association with music for sleep. Similarly, understanding of particular terms could differ between individuals, for example due to language differences.

Asking participants to rate only a subset of the total music samples was done in order to improve convenience of the survey. However, this may have introduced certain biases for ratings of individual tracks. Likewise, the randomisation of stimuli presented led to a degree of variance in the number of ratings received by each track.

Only a sample of our participants completed the study in the evening. Because our primary interest is music that can be used for sleep, this is a limitation, as time of day can affect not only mood but also attention and vigilance [58], and potentially participants' experience of the music. Furthermore, our results are indications of perceptions in a listening study and not in the context of real sleep at night-time. The translation of these factors to the real efficacy of music for sleep remains to be tested.

## Implications

Recent technological developments have coincided with a proliferation in mental health applications and digital therapeutics [59,60]. Personalisation is a key challenge for this area. Programs are being developed that use music to promote wellbeing [61,62] and many sleep apps include some form of soundscaping to help users drift off. However, these tend to be generalised or rely on musical features-based protocols. Our research reveals that such an approach may be limited and further optimisation requires a better incorporation of subjective individual feedback. For example, prior assessments of a selection of music might incorporate subjective measures to fine-tune a sample selection for an individual.

Theoretically speaking, our work has situated our understanding of the use of music for sleep in a wider realm of affective and emotional responses to music. This particularly relates to valence and arousal associations, providing empirical validation for theories previously put forward on the mechanisms related to how music can help with sleep [20]. Additionally, we have identified further affordances associated with wellbeing such as comfort and absorption that play a role, and offer suggestions towards teasing apart the nature of dissociation. This opens the door to further investigations into the contributing factors, such as what makes music comforting and the potential role of social surrogacy. By integrating subjective factors and musical features in our analysis we have taken a novel approach to conceptualising sleep music that reveals and informs gaps in the field. Although the suggestion of individual variability is not new [10,11,32], we have empirically shown the complexity in the interplay between musical features and subjective appraisals that need to be better understood in music for sleep.

## Conclusion

The subjective nature of musical experiences is routinely discussed in music psychology research. Our results establish this with respect to music for sleep where previously emphasis has focused on music type and objective characteristics. Musical features still play a role, but there is clear potential for subjective optimisation to improve what music is used in sleep therapies. We have provided a conceptual foundation and a basic ranking of pieces considered to be sleep inducing, as rated by 108 participants, with the results supporting the selection of specific pieces used in previous studies [20,47]. The results provide valuable insight into the types of subjective evaluations relevant for sleep music and a foundation for more specific probing. In particular, notions of comfort and freeing of the mind are important for music perceived to promote sleep.

## Supporting information

**S1 Table. Music selection.**
(DOCX)

**S2 Table. Additional participant information.**
(DOCX)

**S3 Table. Further statistical results.**
(DOCX)

**S4 Table. List and description of ten most sleep-inducing pieces.**
(DOCX)

## Acknowledgments

We would like to thank the students who contributed their compositions which were used as stimuli in this study, and Professor Dorothy Ker for her support in organising this collaboration.

## Author contributions

**Conceptualization:** Rory Kirk, Maan van de Werken, Renee Timmers.

**Data curation:** Rory Kirk.

**Formal analysis:** Rory Kirk.

**Investigation:** Rory Kirk.

**Methodology:** Rory Kirk, Maan van de Werken, Renee Timmers.

**Project administration:** Renee Timmers.

**Supervision:** George Panoutsos, Maan van de Werken, Renee Timmers.

**Validation:** Rory Kirk, George Panoutsos, Maan van de Werken, Renee Timmers.

**Visualization:** Rory Kirk.

**Writing – original draft:** Rory Kirk, Renee Timmers.

**Writing – review & editing:** Rory Kirk, George Panoutsos, Maan van de Werken, Renee Timmers.

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
