## [Decision Letter · Decision Letter 0]

25 Feb 2025

Dear Dr. Kirk,

Thank you for submitting your manuscript to PLOS ONE. After careful consideration, we feel that it has merit but does not fully meet PLOS ONE’s publication criteria as it currently stands. Therefore, we invite you to submit a revised version of the manuscript that addresses the points raised during the review process.

Two experts in the field raised constructive points that would further improve the readership of the manuscript. Please find one review attached in PDF format. Address all raised concerns in a point-by-point fashion in the rebuttal.

We look forward to receiving your revised manuscript.

Kind regards,

Seung-Goo Kim, Ph.D.

Academic Editor

PLOS ONE

Journal Requirements:

Reviewers' comments:

Reviewer's Responses to Questions

**Comments to the Author**

1. Is the manuscript technically sound, and do the data support the conclusions?

Reviewer #1: Yes

Reviewer #2: Yes

2. Has the statistical analysis been performed appropriately and rigorously?

Reviewer #1: I Don't Know

Reviewer #2: Yes

3. Have the authors made all data underlying the findings in their manuscript fully available?

Reviewer #1: Yes

Reviewer #2: Yes

4. Is the manuscript presented in an intelligible fashion and written in standard English?

Reviewer #1: Yes

Reviewer #2: Yes

Reviewer #1: Well-structured, informative, and generally well-written comprehensive overview of the role of music in aiding sleep, particularly by exploring the subjective perceptions of the listener and the objective characteristics of the music that may contribute to its sleep-inducing qualities. These approaches combine subjective experiences with objective musical analysis, allowing us to understand how music can be used as a therapeutic tool to improve sleep. The presents organized sections, including an abstract, introduction, and detailed discussions on theoretical frameworks, methodologies, and results. Specifically:

i. Abstract:

The abstract concisely summarizes the study’s objectives, methods, and results, providing a quick understanding of the purpose and scope. It offers an overview of research on the role of music in promoting sleep and mentions key factors considered, such as the impact of musical characteristics on the listener’s sleep quality. A more explicit mention of the study’s specific perspectives could help further distinguish this work from previous similar research.

ii. Introduction:

The introduction provides context on the increasing prevalence of sleep disorders in modern society and highlights the potential of music as an intervention to improve sleep. It paves the way for the research by describing the historical use of music as a sleep aid and establishing the justification for scientifically studying this relationship. The research question is clearly presented, and the importance of understanding both subjective and objective factors involved in the effectiveness of music for sleep is emphasized. It could be interesting to elaborate on how existing research gaps are addressed by this approach, particularly regarding the integration of listeners’ subjective experiences and objective analysis of musical characteristics.

iii. Study Design:

The study design is clearly presented, detailing the methods used to explore how listeners' subjective perceptions and the objective characteristics of music contribute to its ability to induce sleep. Including self-assessments from listeners and objective analyses of musical elements such as tempo, harmony, and rhythm is a strength, as it provides a holistic understanding of how music works as a sleep aid. More detailed explanations of the participants' demographic characteristics and parameters would enhance the study’s generalizability. Information on participants' culture, age ranges, gender distribution, and sleep habits allows readers to assess whether the results are applicable to a broader population and how, or if specific subgroups are more likely to benefit from music for sleep.

iv. Psychological Factors:

The discussion of psychological mechanisms influencing sleep, including wakefulness, emotion, and relaxation, is a strong point. By highlighting how music can regulate these factors, the text it connects the physiological and psychological aspects of sleep to music itself. It successfully explains how specific musical elements can influence the listener's emotional state, which in turn affects sleep onset and quality. This section could benefit from a deeper exploration of the neurobiological mechanisms in such an interaction, particularly how music interacts with brainwave patterns, waves, and autonomic nervous system activity during sleep. Further reflection on how music may affect different stages of sleep, such as REM and non-REM sleep, could also enrich the analysis.

v. Theoretical Frameworks:

The interdisciplinary approach draws on a variety of theoretical frameworks from music psychology, sleep research, and cognitive neuroscience, providing a flexible, broad, and comprehensive theoretical foundation. References to previous research on the effects of music on sleep are well integrated, linking this study to the existing literature. The work can benefit from a more explicit discussion on how this work builds upon or modifies previous theories. How does the study's focus on listeners' subjective perceptions contribute to or challenge established models of music’s impact on sleep, such as the "relaxation response theory" or the "entrainment theory"?

vi. Acronyms:

Some readers may not be familiar with terms like "MSSD" (Music Streaming Session Dataset), which could make the reading less accessible. The text could benefit from a very brief explanation or development when these terms are introduced for the first time (or even a later reminder - for example - for very specific citations/references). A dataset generated by users on musical preferences and listening habits related to sleep. In addition to any post footnotes or glossary appendixing, occasionally recalling/clarifying acronyms is a simple yet effective way to ensure accessibility for a broader, audience.

vii. Sentence Length and Structure:

Some sections contain long, complex sentences that may be difficult to follow for certain readers, particularly in the theory and result sections. Breaking these sentences into shorter, more digestible chunks would significantly improve the manuscript’s readability. When discussing specific theories or findings, it may be helpful to present each idea or concept separately, concisely, and in short sentences rather than grouping multiple ideas into one dense statement.

viii. Flow and Transitions:

Transitions between different sections, such as those between psychological theories and specific findings, present a challenge in the straightforward narration of the information we construct. A special revision is needed to maintain the logical flow of ideas and focus of attention, allowing readers to follow the narrative more easily. For example, after analyzing the psychological mechanisms involved in sleep, the results could more explicitly link these theoretical concepts by drawing on both ideas and expanding observed findings to strengthen the ties between theory and practice while creating a clearer narrative for the reader.

ix. Methodological Clarity:

There is still room to improve the methodology by providing more details, as it would make it more transparent to allow readers to assess study’s validity and applicability. Any additional information on participant recruitment is always welcome, and possible tendencies or unavoidable filters, as it would help readers understand the representativeness of the sample, and promote the sharing of these results and analyses ad hoc. The music selection process could be more detailed: was a specific music genre chosen, and how/why? How was the selection based on listener preferences?

x. Musical Diversity and Impact:

An important consideration that could be addressed in the text is whether musical diversity, in terms of genre or cultural context and familiarity, was taken into account in the study’s design. The text could examine whether different genres (e.g., classical, ambient, jazz) or music from different cultures (listeners’ or others’) may have different effects on sleep. For example, how might an audience’s cultural context influence their perception of music from different origins and its effectiveness in inducing sleep? How can we measure or report such impressions? This seems an important emerging question, particularly in a globally interconnected world where people have access to a vast variety of music, a world that was previously extremely vast.

xi. Highlighting Novelty and Contributions:

The text summarizes existing literature on music and sleep but could explicitly highlight new/relevant/similar aspects based on previous research directions. Integrating listeners’ subjective perceptions alongside objective musical analysis can be a unique contribution to the field, which could be further emphasized in both the abstract and the introduction, as mentioned in the specific comments for such points, highlighting how this study fills a gap in current research and advances our understanding of how listener customization and preferences influence sleep outcomes.

xii. Practical Applications:

A more in-depth discussion on the practical applications of the results would enhance the findings and clarify how the results could inform the development of personalized sleep music technologies, compositions, music applications, or therapeutic interventions. How could sleep music recommendations be personalized based on each listener’s preferences, emotional responses, or sleep habits? What practical measures can music therapists, app developers, or healthcare providers take to integrate these results into concrete applications?

The work contributes to the understanding of human sleep patterns and ways by exploring the interplay of subjective listener perceptions and objective musical features during sleep. It adds to the growing body of research on the use of music for health and well-being, particularly in the realm of sleep. The study's comprehensive design and integration of both psychological and musical elements provide a strong foundation for understanding how music can be used as a tool to improve sleep quality. The potential impact can be improoved if methodological ambiguities, dense writing, and a lack of emphasis on practical applications are better adressed through a clear explanation on to better focus upon novelty and the expanded discussion of practical usage. Therefore, the text could be further strengthened enhancing clarity, accessibility, and relevance, particularly in applied contexts like therapeutic music composition, music apps for sleep, and consumer-focused sleep aids.

Reviewer #2: Overall the authors present a study with a clear purpose, well executed while being aware of possible limitations and choices made for reasons of feasibility. The paper is well written and is supported by strong methodology. More comments in attachment

**Do you want your identity to be public for this peer review?** For information about this choice, including consent withdrawal, please see our Privacy Policy

Reviewer #1: **Yes: ** Jesús Aparicio de Soto

Reviewer #2: **Yes: ** Rebecca Jane Scarratt

---

## [Author Response · Author response to Decision Letter 1]

11 Jul 2025

Reviewer(s)' Comments to Author:

Reviewer #1:

Well-structured, informative, and generally well-written comprehensive overview of the role of music in aiding sleep, particularly by exploring the subjective perceptions of the listener and the objective characteristics of the music that may contribute to its sleep-inducing qualities. These approaches combine subjective experiences with objective musical analysis, allowing us to understand how music can be used as a therapeutic tool to improve sleep. The presents organized sections, including an abstract, introduction, and detailed discussions on theoretical frameworks, methodologies, and results. Specifically:

i. Abstract:

The abstract concisely summarizes the study’s objectives, methods, and results, providing a quick understanding of the purpose and scope. It offers an overview of research on the role of music in promoting sleep and mentions key factors considered, such as the impact of musical characteristics on the listener’s sleep quality. A more explicit mention of the study’s specific perspectives could help further distinguish this work from previous similar research.

Author response: Thank you very much for this suggestion. We have added a couple of sentences to the abstract to emphasise this point (lines 34-38).

ii. Introduction:

The introduction provides context on the increasing prevalence of sleep disorders in modern society and highlights the potential of music as an intervention to improve sleep. It paves the way for the research by describing the historical use of music as a sleep aid and establishing the justification for scientifically studying this relationship. The research question is clearly presented, and the importance of understanding both subjective and objective factors involved in the effectiveness of music for sleep is emphasized. It could be interesting to elaborate on how existing research gaps are addressed by this approach, particularly regarding the integration of listeners’ subjective experiences and objective analysis of musical characteristics.

Author response: We have made small edits throughout the Introduction to emphasise the advantages of integrating subjective and objective analysis, and expanded the Current study section (lines 239-264) to better pinpoint our addressing of this gap (and address other comments by Reviewer #2).

iii. Study Design:

The study design is clearly presented, detailing the methods used to explore how listeners' subjective perceptions and the objective characteristics of music contribute to its ability to induce sleep. Including self-assessments from listeners and objective analyses of musical elements such as tempo, harmony, and rhythm is a strength, as it provides a holistic understanding of how music works as a sleep aid. More detailed explanations of the participants' demographic characteristics and parameters would enhance the study’s generalizability. Information on participants' culture, age ranges, gender distribution, and sleep habits allows readers to assess whether the results are applicable to a broader population and how, or if specific subgroups are more likely to benefit from music for sleep.

Author response: We have added details to the Methods and Materials section to explain the types of background information we gathered in this study (lines 345-353) and provided more details on our sample in the Results section (lines 471-478). We have also added descriptive statistics for all of our background measures to the Supplementary Material (see S2).

iv. Psychological Factors:

The discussion of psychological mechanisms influencing sleep, including wakefulness, emotion, and relaxation, is a strong point. By highlighting how music can regulate these factors, the text it connects the physiological and psychological aspects of sleep to music itself. It successfully explains how specific musical elements can influence the listener's emotional state, which in turn affects sleep onset and quality. This section could benefit from a deeper exploration of the neurobiological mechanisms in such an interaction, particularly how music interacts with brainwave patterns, waves, and autonomic nervous system activity during sleep. Further reflection on how music may affect different stages of sleep, such as REM and non-REM sleep, could also enrich the analysis.

Author response: Thank you for this suggestion. We appreciate that there is a dearth of research in the area of music and neuroscience, and indeed neural activity is a fundamental indicator of sleep behaviour. However, as the topic of this paper is not neuroscientific we feel that our literature review should be focused on aspects more specifically relevant to the theories and measures incorporated in the study.

v. Theoretical Frameworks:

The interdisciplinary approach draws on a variety of theoretical frameworks from music psychology, sleep research, and cognitive neuroscience, providing a flexible, broad, and comprehensive theoretical foundation. References to previous research on the effects of music on sleep are well integrated, linking this study to the existing literature. The work can benefit from a more explicit discussion on how this work builds upon or modifies previous theories. How does the study's focus on listeners' subjective perceptions contribute to or challenge established models of music’s impact on sleep, such as the "relaxation response theory" or the "entrainment theory"?

Author response: We have elaborated in the Discussion to discuss the contributions of this research to theories related to the use of music for sleep, and concepts related to music and emotions and the use of music for the purpose of wellbeing.

vi. Acronyms:

Some readers may not be familiar with terms like "MSSD" (Music Streaming Session Dataset), which could make the reading less accessible. The text could benefit from a very brief explanation or development when these terms are introduced for the first time (or even a later reminder - for example - for very specific citations/references). A dataset generated by users on musical preferences and listening habits related to sleep. In addition to any post footnotes or glossary appendixing, occasionally recalling/clarifying acronyms is a simple yet effective way to ensure accessibility for a broader, audience.

Author response: Thank you for pointing this out, as we only refer to the MSSD once we have removed this acronym and added details to our description of the source for clarity (lines 184-185). We have checked the document for any other use of acronyms and made similar amendments.

vii. Sentence Length and Structure:

Some sections contain long, complex sentences that may be difficult to follow for certain readers, particularly in the theory and result sections. Breaking these sentences into shorter, more digestible chunks would significantly improve the manuscript’s readability. When discussing specific theories or findings, it may be helpful to present each idea or concept separately, concisely, and in short sentences rather than grouping multiple ideas into one dense statement.

Author response: We have proofread the manuscript and made edits to improve the sentence structure and general flow of the document.

viii. Flow and Transitions:

Transitions between different sections, such as those between psychological theories and specific findings, present a challenge in the straightforward narration of the information we construct. A special revision is needed to maintain the logical flow of ideas and focus of attention, allowing readers to follow the narrative more easily. For example, after analyzing the psychological mechanisms involved in sleep, the results could more explicitly link these theoretical concepts by drawing on both ideas and expanding observed findings to strengthen the ties between theory and practice while creating a clearer narrative for the reader.

Author response: We have revised the article to improve the flow of language for the reader.

ix. Methodological Clarity:

There is still room to improve the methodology by providing more details, as it would make it more transparent to allow readers to assess study’s validity and applicability. Any additional information on participant recruitment is always welcome, and possible tendencies or unavoidable filters, as it would help readers understand the representativeness of the sample, and promote the sharing of these results and analyses ad hoc. The music selection process could be more detailed: was a specific music genre chosen, and how/why? How was the selection based on listener preferences?

Author response: We have added additional details to describe our sample demographics (lines 471-478), and used our Discussion to elaborate on our design limitations related to our participants and recruitment (lines 889-903). A detailed description of the music selection process is provided in the supplementary material, however we have also added details to the Stimuli part of the Methods and Materials section (line 273). We deliberately did not choose any specific genre with the ideal of gathering a mixed selection of music that broadly defines each category of music.

x. Musical Diversity and Impact:

An important consideration that could be addressed in the text is whether musical diversity, in terms of genre or cultural context and familiarity, was taken into account in the study’s design. The text could examine whether different genres (e.g., classical, ambient, jazz) or music from different cultures (listeners’ or others’) may have different effects on sleep. For example, how might an audience’s cultural context influence their perception of music from different origins and its effectiveness in inducing sleep? How can we measure or report such impressions? This seems an important emerging question, particularly in a globally interconnected world where people have access to a vast variety of music, a world that was previously extremely vast.

Author response: This is a very important limitation in our study and we greatly appreciate this point. As regards previous comments, we have tried to include more information on our sample in our reporting and in the Discussion section we have added comments to our sample limitations (lines 899-903). Specifically, we have mentioned the cultural significance of music for sleep that is evident in the prevalence of lullabies, and the open question of how this diverges into adulthood. A full investigation of individual differences was outside of the scope of this study, indeed our sample was notably skewed in several demographics. Ultimately, a proper evaluation of cross-cultural differences and other demographic variables would require a study with a different design, or much larger sample, with systematic consideration of stimuli, inclusion criteria and recruitment strategy (such as translation of the survey). This is something that we hope to pursue in future.

xi. Highlighting Novelty and Contributions:

The text summarizes existing literature on music and sleep but could explicitly highlight new/relevant/similar aspects based on previous research directions. Integrating listeners’ subjective perceptions alongside objective musical analysis can be a unique contribution to the field, which could be further emphasized in both the abstract and the introduction, as mentioned in the specific comments for such points, highlighting how this study fills a gap in current research and advances our understanding of how listener customization and preferences influence sleep outcomes.

Author response: Thank you for this reflection. We have made edits to emphasise this point throughout, and make a more explicit statement in the Implications section of our Discussion about the novelty of our approach and its contribution to the topic (lines 922-943).

xii. Practical Applications:

A more in-depth discussion on the practical applications of the results would enhance the findings and clarify how the results could inform the development of personalized sleep music technologies, compositions, music applications, or therapeutic interventions. How could sleep music recommendations be personalized based on each listener’s preferences, emotional responses, or sleep habits? What practical measures can music therapists, app developers, or healthcare providers take to integrate these results into concrete applications?

Author response: Alongside highlighting novelty and contributions (above) we also discuss the practical and theoretical implications of our research in the Implications section of our Discussion (lines 922-943).

The work contributes to the understanding of human sleep patterns and ways by exploring the interplay of subjective listener perceptions and objective musical features during sleep. It adds to the growing body of research on the use of music for health and well-being, particularly in the realm of sleep. The study's comprehensive design and integration of both psychological and musical elements provide a strong foundation for understanding how music can be used as a tool to improve sleep quality. The potential impact can be improoved if methodological ambiguities, dense writing, and a lack of emphasis on practical applications are better adressed through a clear explanation on to better focus upon novelty and the expanded discussion of practical usage. Therefore, the text could be further strengthened enhancing clarity, accessibility, and relevance, particularly in applied contexts like therapeutic music composition, music apps for sleep, and consumer-focused sleep aids.

Author response: Thank you very much for your feedback. We hope that we have addressed your concerns, and appreciate any further comments.

Reviewer #2:

Overall the authors present a study with a clear purpose, well executed while being aware of possible limitations and choices made for reasons of feasibility. The paper is well written and is supported by strong methodology.

My main question is the inclusion of the scale Distraction-Freeing the mind and the use of distraction as negative. I understand the complexity of the word distraction in that it could cover both negative distraction and positive distraction. For example, a track could distract from concentration and have a negative impact, as was meant with the inclusion of this scale. However, how about the case when individuals would use music to mask external sounds? Then distraction would be necessary and might have a positive effect on sleep. It would not be considered as Freeing the mind. Perhaps a scale from ‘Over-stimulation to Under-stimulation' of music would be more appropriate, as music could distract positively in cases of appropriate stimulation but negatively in cases of over-stimulation.

Author response: We indeed use distraction in this context as negative, looking at it from the perspective of the music being a distraction (or attention grabbing). This use relates to our focus on characterising the subjective properties of the music as experienced by the participant rather than the use of the music (see also response to your next comment). We have added a point in the Discussion that acknowledges that the questionnaire generally and poles used in the items require further validation (lines 904-911).

Furthermore, three reasons for using sleep are mentioned (arousal, valence and distraction). However, there are other possible reasons why individuals turn to music for sleep (such as masking) as proposed by Dickson & Schubert 2020 and Trahan et al. 2018. It would be good to see these mentioned with an explanation of why you are not taking these into consideration for this study.

Author response: Thank you for this point, this is indeed something we had not explicitly acknowledged. In our study we were interested in characterising the subjective properties or affordances of music, which we would distinguish from characterising the uses of music for sleep. There is of course a lot of overlap; the reasons for using music suggest what affordances music may have, and indeed Dickson & Schubert (2020) and Trahan et al. (2018) both refer to mood and distraction in their studies, for example. But we think there’s an impo

---

## [Decision Letter · Decision Letter 1]

30 Jul 2025

The relative contributions of subjective and musical factors in music for sleep

PONE-D-24-49835R1

Dear Dr. Kirk,

We’re pleased to inform you that your manuscript has been judged scientifically suitable for publication and will be formally accepted for publication once it meets all outstanding technical requirements.

Kind regards,

Seung-Goo Kim, Ph.D.

Academic Editor

PLOS ONE

Additional Editor Comments (optional):

Reviewers' comments:

Reviewer's Responses to Questions

**Comments to the Author**

Reviewer #2: (No Response)

2. Is the manuscript technically sound, and do the data support the conclusions?

Reviewer #2: (No Response)

3. Has the statistical analysis been performed appropriately and rigorously?

Reviewer #2: (No Response)

4. Have the authors made all data underlying the findings in their manuscript fully available?

Reviewer #2: (No Response)

5. Is the manuscript presented in an intelligible fashion and written in standard English?

Reviewer #2: (No Response)

Reviewer #2: (No Response)

**Do you want your identity to be public for this peer review?** For information about this choice, including consent withdrawal, please see our Privacy Policy

Reviewer #2: **Yes: ** Rebecca Jane Scarratt

---

## [Editor Report · Acceptance letter]

PONE-D-24-49835R1

PLOS ONE

Dear Dr. Kirk,

I'm pleased to inform you that your manuscript has been deemed suitable for publication in PLOS ONE. Congratulations! Your manuscript is now being handed over to our production team.

Kind regards,

on behalf of

Dr. Seung-Goo Kim

Academic Editor

PLOS ONE